# Adverse Childhood Experiences (ACEs) Screening in Pediatric Primary Care: Is “Social Drivers of Health (SDoH) Screening” Sufficient?

**DOI:** 10.3390/ijerph22111644

**Published:** 2025-10-29

**Authors:** Sylvia Zielinski, Jocelyn Valdez, Juliana James, Jennifer Gates, Bhavik Patel, Tre DeVon Gissandaner, Rachel Feurstein, Ryan Levy, Wanda Vargas, Evelyn Berger-Jenkins

**Affiliations:** 1New York-Presbyterian, Columbia University Irving Medical Center, New York, NY 10032, USA; eb283@cumc.columbia.edu; 2Mailman School of Public Health, Columbia University, New York, NY 10032, USA; jvaldez1@health.nyc.gov (J.V.); jj3287@cumc.columbia.edu (J.J.); bhavik.patel@atria.org (B.P.); 3New York State Psychiatric Institute, Columbia University, New York, NY 10032, USA; 4School of Medicine, University of Missouri-Kansas City, Kansas City, MO 64108, USA; 5Division of Population and Community Health, New York-Presbyterian, New York, NY 10032, USA; qqm9002@nyp.org (R.F.); ryl9010@nyp.org (R.L.); wav9004@nyp.org (W.V.)

**Keywords:** ACES, SDOH, trauma, screening, pediatric primary care

## Abstract

Adverse childhood experiences (ACEs) are established predictors of long-term health risks. While pediatric practices increasingly screen for social drivers of health (SDOH) and other family psycho-social stressors, routine ACEs screening is not recommended due to lack of evidence for long-term benefit and concerns over stigmatization, re-traumatization, and non-standardized follow-up protocols. We piloted routine ACEs screening in Pediatric Primary Care practices that already routinely screen for SDOH, maternal depression and intimate partner violence (IPV). This retrospective chart review (2016–2020) explored the extent to which these family psycho-social screenings could serve as a relative proxy for ACEs identification. Among 1492 participants (738 children aged 0–5 and 690 caregivers mean age 30.3 ± 6.9), ACE and SDOH screening results were significantly associated (*p* < 0.002), particularly with housing insecurity (*p* < 0.014). However, 51.7% of individuals who reported a positive ACE screen were not flagged by the SDOH measure (false negatives), indicating relatively poor sensitivity. The negative predictive value for negative SDOH screens and negative ACEs was higher at 86%. These findings suggest that SDOH screening misses over half of true positives, and therefore reliance on SDOH screening alone may underestimate ACE exposure in pediatric primary care.

## 1. Introduction

Adverse childhood experiences (ACEs), defined by Feletti et al. in 1998 as abuse, neglect, and household dysfunction categorized into 10 exposures, have been associated with long-term health consequences [1]. Various longitudinal studies have shown that dose-dependent exposure to ACEs is associated with a range of negative health outcomes, including chronic conditions like cardiovascular disease, mental health disorders, and increased engagement in health risk behaviors [2,3,4]. In a review of prevalence by Swedo et al. in 2023, 63.9% of adults in the United States were found to have experienced at least one ACE, and 17.3% reported four or more ACEs by age 18 [5]. Parental reports from the National Survey of Children’s Health indicate that over one-third of children ages 0–5 experience at least one ACE [6]. Notably, there are significant disparities in both exposure and negative outcomes related to ACEs, particularly with those in lower-income households and lower educational attainment [5,7,8]. Racial disparities are demonstrated in studies that found higher mean scores in Black, Hispanic, and American Indian/Alaska Native populations when compared to their White and Asian counterparts [9,10].

Many primary care pediatric practices screen for family-level psycho-social stressors such as social drivers of health (SDOH), maternal depression, and intimate partner violence (IPV), for example. These family stressors contribute significantly to trauma; however, there is current debate over whether screening for child or caregiver ACEs themselves, in addition to these stressors, is an appropriate and recommended practice in primary care settings [11]. The American Academy of Pediatrics does not currently recommend universal screening for ACEs, based on gaps in research and lack of evidence that formal screening improves long-term health outcomes [12]. When considering screening implementation, there have also been concerns raised regarding fear of stigmatization, re-traumatization, discomfort of caregivers, lack of evidence-based protocols for positive screens, and mandated reporting for providers [12,13]. Given that the ACEs screen was developed as a research tool and not a screening tool, and that the symptoms as well as impact from trauma vary from person to person, the current recommendation is to screen for symptoms as opposed to exposure to traumatic experiences as a form of tertiary prevention [12]. Researchers have called for additional research on the settings and circumstances in which ACEs screening may be justified, and for alternative screenings or tools that may be used for early identification of trauma in lieu of ACEs screens.

Although evidence for long-term benefits of screening for ACEs is lacking, there is continued interest in addressing ACEs and trauma in healthcare and public health settings, given the strong correlation of ACEs with poor health outcomes. The current literature piloting modified ACEs screening in pediatric primary care has used items from existing tools such as the Parent Screening Questionnaire (PSQ) and the Children’s HealthWatch survey. These studies show that screening improves identification of adversity and may increase referrals to services [14]. Studies have further shown that screening paired with a psychosocial assessment and warm handoff to behavioral health services has been associated with an increased likelihood of a behavioral health visit within 90 days and overall increase in social work referrals [15,16].

Recognizing what current screening tools are routinely implemented in primary care settings, some may consider other family psycho-social screens including social drivers of health, maternal depression, and intimate partner violence screening as a “proxy” for ACEs, as there is a strong association between these environmental factors. Multiple systematic reviews have demonstrated an increased risk of ACEs in patients exposed to low parental education, poverty, and housing instability [7,17,18,19,20]. However, the extent to which SDOH or other family psycho-social stressor screening alone, without ACEs, effectively identifies ACEs exposure is not clear. Therefore, in order to understand the comparability of ACEs screening instruments to alternatives, we compared screening data from pediatric settings that routinely administer screening for family psycho-social stressors, including SDOH, and added ACEs screening that was implemented as part of an outreach effort of our hospital’s trauma treatment program.

## 2. Materials and Methods

### 2.1. Study Design and Participants

This study included a retrospective chart review of a ~ three-year-long (from November 2016 to March 2020) ACEs screening pilot program across four Pediatric Primary Care settings. As part of routine care for the last 10 years, caregivers complete the family psycho-social screens mentioned above (SDOH, maternal depression, and intimate partner violence) in the waiting room on an annual basis when they present for their child’s wellness visits. All screenings completed in the study were available in either English or Spanish, based on the family’s preference. The SDOH screen is adopted from measures used by the Centers for Medicare & Medicaid Services, and includes items assessing food insecurity, housing insecurity, utility needs, and transportation concerns; see Figure 1 [21]. The SDOH screen is completed by the caregiver in reference to the family unit. The screen is completely self-administered on electronic tablets that are distributed to caregivers upon registration and is part of routine practice. For this study, each insecurity in the SDOH screen was scored as a dichotomous (yes/no) response. Similarly, the well-established Patient Health Questionnaire (PHQ-9) and HITS (Hurt, Insulted, Threatened and Screamed) screens were used to assess caregiver depression and intimate partner violence, respectively. These were also scored according to previously established, validated and published standards [22,23]. See further description of data collection and analysis below.

Our hospital contains a trauma treatment program that specializes in holistic treatment of families affected by trauma, including psychotherapy, art therapy, case management, legal and spiritual support. The center is staffed largely by psychology, social work, and psychiatry clinicians, among other support staff. As part of an outreach effort of the trauma treatment center, psychologists spent three years on ACEs screening and response to screening efforts within the primary care setting. They reviewed wellness visit schedules and approached caregivers of children ages 0–5 years who presented for wellness child visits. Caregivers were asked to complete both child and caregiver ACEs in addition to the other routine screens upon registration for their visit. The original 10-item ACE screen from the CDC-Kaiser Permanente study was used, as shown in Appendix A [1]. It was completed on paper, and self-administered by parents who were asked to simply provide the total number of ACEs they (or their child) had been exposed to [24]. The ACE screen was scored according to standard cutoffs, with 4 or more ACEs being considered “positive” in caregivers, and 1 or more ACEs being considered “positive” in children. Team members of the hospital’s trauma treatment program scored the ACEs in real-time, and provided pediatricians with the results as well as referral information to on-site clinicians who conducted more detailed assessment and management as needed. Management of the other routine screens (SDOH, maternal depression and intimate partner violence) was not affected by this study, and providers referred for services as they had routinely performed prior to the study and with the use of on-site social workers as well as community health workers when needed and at their discretion.

Overall, 1492 participants (children and caregivers) were analyzed with inclusion criteria of having a child between the ages of 0–5 years old and having a well visit scheduled. Exclusion criteria included families with a child outside of the age range for our hospital’s trauma treatment program (0–5 years). Families who had incomplete or missing data were removed (N = 64). If families had more than one child entry, only the oldest child was included. Ultimately, 1428 participants (738 children and 690 caregivers) were included in the study. The mismatch in the number of caregivers and children is mostly related to missing data, primarily occurring for caregivers.

This study was approved by the Institutional Review Board of Columbia University.

### 2.2. Data Collection

Child and caregiver demographics (age, sex and self-reported race/ethnicity, primary language) and routine family psycho-social screening (SDOH, maternal depression and intimate partner violence) data were extracted from patient medical charts by our hospital’s IT team.

ACEs data were manually entered into a database and cross-checked by at least two team members for accuracy. Missing data was reviewed by pulling the paper ACEs when possible and manually reviewing responses. Participants were excluded when missing data could not be found (N = 64, 4%). All child and caregiver ACEs were entered as a total raw score, and then coded to represent a composite “positive/negative” dichotomous result based on previously established cutoffs of ≥1 considered positive for children, and ≥4 considered positive for caregivers.

SDOH screens were similarly coded as dichotomous “positive/negative” results for each of the four SDOH (food insecurity, housing, transportation and utilities) included in the screen. We also created a composite “positive/negative” SDOH score that was considered positive if at least one SDOH was reported. Not all participants who completed the ACEs screening also completed the SDOH questionnaire. Those participants with no SDOH entries were included for demographic and ACEs frequency analyses and excluded when bivariate analyses including SDOH were calculated.

### 2.3. Data Analysis

All data were analyzed using SAS Studio 3.82 software.

Descriptive statistics (including frequencies, means, and percentages) were calculated to characterize the populations’ demographic data. SDOH and ACEs raw and dichotomous (positive-negative) scores were also calculated. Bivariate analyses, using chi-square or Fisher’s exact tests, were used to assess for associations between the SDOH and ACEs screens. Similar analyses were conducted for maternal depression and intimate partner violence compared to ACEs screens. All calculations were completed for child and caregiver participants separately. False positive (% of positive SDOH with negative ACEs) and false negative (% of negative SDOH with positive ACEs) rates were calculated using a PROC SQL procedure to identify cases that might be erroneously classified, reflecting potential Type I or Type II error. Odds ratios (OR) with 95% confidence intervals were calculated to evaluate the relative likelihood of ACE positivity between SDOH-positive and SDOH-negative groups. Misclassification rates and OR were not calculated for maternal depression or IPV due to relatively small numbers of responses on these screens compared to SDOH screens, as well as the fact that SDOH is a more widespread measure of family stress that is used routinely in pediatric primary care settings. Sensitivity calculations were conducted to investigate for any dose-dependent relationship between ACE exposure and positive SDOH screening.

Significance was defined as a *p* value less than 0.05.

## 3. Results

### 3.1. Demographics

Of the total 1428 participants screened, 738 were children and 690 were caregivers. Amongst the child participants, there was an even distribution of males and females. Of the caregivers, data about their sex was mostly missing. The average age of participants was 1.6 years old for the children, and 30.3 years for caregivers. Approximately 78.1% of participants were Latinx and 15.7% were Black or African American. The primary language for the majority of patients was English. See Table 1 for further descriptives.

### 3.2. Participant Responses to Screeners

Of the 1428 participants, 212 (14.8%) children and caregivers screened positive on the 10-item ACEs questionnaire. As seen in Table 2, 18.7% of children screened positive for one or more ACE, while 10.7% of caregivers endorsed four or more ACEs. Of those who completed the SDOH screen (N = 838), 310 (37.0%) screened positive for having experienced at least one SDOH. When analyzed by individual SDOH items, 40 (5.0%) reported utilities concerns, 84 (10.4%) reported transportation concerns, 175 (21.8%) reported housing insecurity and 167 (20.5%) reported food insecurity. Fewer families (36, 4.6%) screened positive for intimate partner violence, and (48, 28.9%) participants endorsed maternal depression.

### 3.3. Bivariate Analyses

There were no statistically significant associations between child ACE and either “any” SDOH or maternal depression screening results in child participants. When analyzing each item of the SDOH screen independently, housing was the only insecurity significantly associated with child ACE positivity (*p* < 0.014). The association between child ACE positivity and reported intimate partner violence also approached significance (*p* < 0.0533) (Table 3).

There was a statistically significant association between caregiver ACE and “any” SDOH (*p* < 0.002). Similarly to child participants, **housing** was the only insecurity significantly associated with caregiver ACE positivity (*p* < 0.003). There were no statistically significant associations between caregiver ACE positivity and reported intimate partner violence or maternal depression (Table 4).

As a whole, the odds of a positive ACEs screen are approximately 2 times higher for those with a positive SDOH screen compared to those with a negative SDOH screen (OR = 1.77, 95% CI: 1.23–2.53).

### 3.4. Misclassification Rates (Table 5)

Despite the increased odds of a positive ACE in participants with reported SDOH and especially housing, we found that the sensitivity for SDOH screening as a proxy for ACEs is still quite low at 48%. This signifies that more than half of the participants who actually have a positive ACE were missed by the SDOH screen. The specificity for SDOH, or the probability that someone who is ACES negative is correctly flagged as negative by the SDOH, is also low at 65%. Positive and negative predictive values of the SDOH screen for detecting ACEs were also assessed. The positive predictive value, defined as the probability that someone flagged positive by SDOH actually is ACES positive, was found to be low at 23%. The only measure of screen accuracy that was relatively high was the negative predictive value. The negative predictive value, or the probability that someone flagged negative by SDOH is truly ACES negative, was 86%, which signifies that negative SDOH screens may be moderately accurate at predicting negative ACEs.
ijerph-22-01644-t005_Table 5Table 5Misclassification rates for dichotomous aces if relying on other psycho-social screens.
PercentageFalse Positive Rate34.59False Negative Rate51.7Sensitivity48.3Specificity65.41Positive Predictive Value22.9Negative Predictive Value85.6

In the sensitivity analysis stratified by group and ACE severity, distinct patterns emerged. Table 6. Among children, sensitivity increased with higher ACE thresholds from 0.44 at ACE of 1 to 0.73 at ACE of 3, while specificity remained relatively stable around 0.64. The decrease in sensitivity seen from ACE of 3 to 4 may be due to a low number of participants who endorsed 4 or more ACEs. The NPV also increased from 0.82 at ACE of 1 to 0.99 at ACE of 3, suggesting that SDOH-negative classification was highly reliable for identifying children with few or no ACEs. However, PPV declined with higher ACE cutoffs from 0.25 to 0.02, which may reflect the smaller number of children with high ACE exposure. Overall accuracy remained moderate, 0.61–0.65.

A similar pattern was observed among caregivers, although changes in sensitivity were more modest, from 0.40 to 0.60 across ACE cutoffs. Specificity again remained stable (0.65), while PPV decreased from 0.63 to 0.20 and NPV increased from 0.42 to 0.92. Accuracy ranged from 0.50 to 0.65.

## 4. Discussion

Adverse childhood experiences are well-established predictors of long-term health risks. However, while pediatric practices increasingly screen for social drivers of health, and other family psycho-social stressors, routine ACEs screening is not recommended due lack of evidence for long-term benefit and concerns over stigmatization, re-traumatization, and the lack of standardized follow-up protocols. Current AAP guidelines emphasize screening for trauma-related symptoms rather than direct exposure, though pilot studies suggest that modified ACEs screening in primary care may improve identification of adversity and linkage to behavioral health services.

In this Pediatric Primary Care population, in a hospital-based practice with access to a trauma treatment program, we were able to successfully screen about 700 families for ACEs in addition to family psycho-social stressors (SDOH, maternal depression and intimate partner violence) over a ~three-year period. The screening was integrated into pre-visit procedures during registration for well visits in the same way other family screening is conducted. We utilized a process similar to other studies and demonstrated to be acceptable to families, with modified ACEs screens in which families simply report the number and not the specific type of ACE exposures. As previous studies demonstrated, our results echoed a high frequency of individuals experiencing ACEs, with 10.7% of caregivers screening positive compared to 17.3% of adults in recent national findings [24]. Our child patient population endorsed lower exposure to ACEs (18.7%) compared to previous reports (33.3%) [6]. Similarly, our population endorsed considerable exposure to SDOHs (40.0% with at least one insecurity including food, housing, transportation or utility insecurities). When analyzing the association of ACEs with SDOH, we found that children or caregivers with a positive SDOH were 1.7 times as likely to have a positive ACE screen compared to those with a negative SDOH. This is consistent with the previous literature that revealed an increased risk of SDOH in adults exposed to ACEs [7,17,18,19,20]. It is not known whether ACEs exposure precedes or is in any way causal in relation to the SDOH found. Many studies have revealed that there is an intergenerational transmission of the effects of ACEs such that downstream physical, social and environmental effects from ACEs may be seen in future generations [25]. In this study, we assessed both caregiver and child ACEs and their associations with SDOH and other family psycho-social stressors reported in cross-section. In particular, we found a significant association between caregiver group ACE positivity and SDOH, specifically housing insecurity. The association between ACEs and both housing insecurity and homelessness has been demonstrated in several prior studies [19,26]. In addition to housing insecurity, we also found an association approaching significance between either child or caregiver ACE positivity and intimate partner violence reported by the caregiver—an association which has also been demonstrated in previous studies [27].

Despite the associations between ACEs and family psycho-social stressors found in both this and previous studies, relying on these screens alone may lead to a poor estimate of the actual extent to which ACEs have occurred, and whether and how they are affecting families. Especially given the existing disparities in ACEs exposure and access to care, these results may provide further evidence for the importance of identifying exposure, symptoms and risk/protective factors in a more precise fashion, avoiding false assumptions or judgements based on existing screens. This type of imprecise assessment might lead to mismatches in care and poor engagement. We demonstrated that most individuals who actually reported a positive ACE screen were not flagged by the SDOH measure (51.70% false negative rate), indicating relatively poor sensitivity. Since the SDOH measure missed the majority of true positives, using such SDOH screens alone might underestimate the prevalence of true ACEs. The fact that the negative predictive value of a negative SDOH screen in predicting negative ACEs was moderately high at 86% is of interest and may be helpful in certain circumstances. Furthermore, when the sensitivity of SDOH screening was stratified by severity of ACE exposure, findings indicated that SDOH status is a moderately sensitive but non-specific indicator of ACE exposure. SDOH appears to better capture individuals with higher levels of adversity, particularly among children, yet its precision decreases at extreme ACE thresholds. The consistently high NPV suggests that SDOH-negative classification reliably identifies those with minimal or no ACE exposure, whereas SDOH-positive classification includes many individuals without high ACE scores.

Limitations to our study included a small sample size and lack of power to detect associations, especially for the maternal depression and intimate partner violence data, which were reported as negative for the majority of participants. These have been known to be underreported in our population of Pediatric Primary Care, and the lack of variance may have limited our ability to detect significant associations with ACEs. Similarly, we acknowledge the apparent inconsistencies in reporting, as some families who reported positive intimate partner violence also reported negative child ACEs when exposure to IPV qualifies as an ACE for children. This study was a naturalistic study that did not assess other variables of interest, including other psycho-social stressors that might mediate or moderate the demonstrated associations with ACEs. Given the naturalistic design, there was intrinsic incomplete or missing data, and an inability to assess which ACEs might be associated with the family psycho-social stressors assessed routinely in our population. Finally, the results of this study may not be generalizable to other populations that are inherently different from our own.

## 5. Conclusions

Adverse childhood experiences are well-established predictors of long-term health risks; however, routine ACEs screening in pediatric primary care settings is not currently recommended. We demonstrated an association between ACE and SDOH screening results, specifically with housing insecurity. Over 50% of individuals who reported a positive ACE screen were not flagged by the SDOH measure, indicating that using such SDOH screens alone may underestimate the prevalence of ACE exposure in primary care settings. Future directions might include the assessment of other routine screens as they relate to ACEs, and the validation of newer trauma screens that assess impact and resilience factors in addition to ACE exposure. This may assist with the implementation of trauma screening in a way that is acceptable to families, feasible within practices, and useful for identifying and managing trauma in primary care settings.

## Figures and Tables

**Figure 1 ijerph-22-01644-f001:**
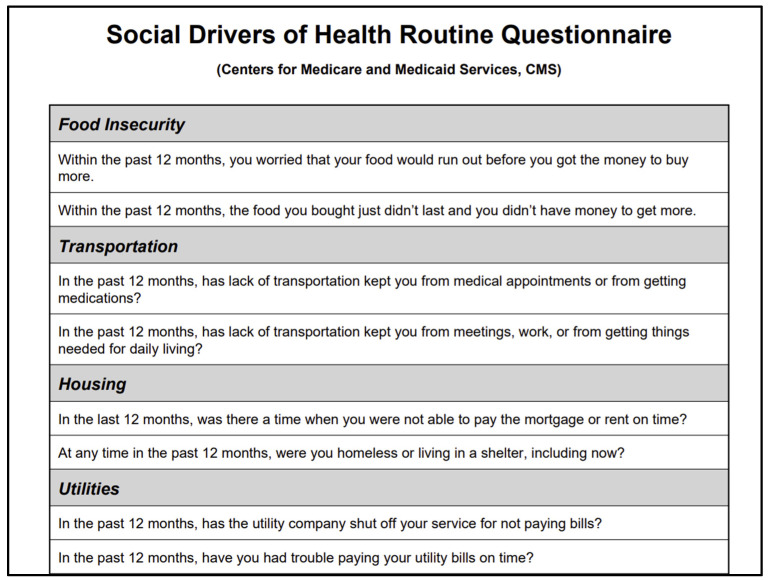
SDOH Screening. (Adapted from the Centers for Medicare & Medicaid Services).

**Table 1 ijerph-22-01644-t001:** Demographics of Study Participants Including Caregivers and Children.

Characteristic	Caregiver	Child	Total
Participants	690	738	1428
Average Age (SD)	30.3 (±6.9)	1.6 (±2.4)	---
Sex (N(%) *)			
Female	43 (91.5)	352 (49.0)	395 (51.6)
Male	4 (8.5)	366 (51.0)	370 (48.4)
Race (N(%))			
Black or African American	10 (26.3)	96 (15.1)	106 (15.7)
Latinx	21 (55.3)	506 (79.4)	527 (78.1)
White	1 (2.6)	14 (2.2)	15 (2.2)
Asian American, Pacific Islander	0	5 (0.8)	5 (0.7)
Other	6 (15.8)	16 (2.5)	22 (3.3)
Missing	29	724	753
Primary Language (N(%))			
English	26 (55.3)	359 (50.1)	385
Spanish	14 (29.8)	317 (44.3)	331
Other	7 (14.9)	40 (5.6)	47
Missing	20	645	665

* NOTE: % is valid percent excluding missing values.

**Table 2 ijerph-22-01644-t002:** Screening Results Including ACEs and Family Psycho-Social Stressor Screens(N(%) *).

	Screen Positive		Screen Negative	
	Caregiver	Child	Caregiver	Child
ACEs	74 (10.7)	138 (18.7)	616 (89.3)	600 (81.3)
SDOHs (any insecurity)	123 (37.8)	187 (36.5)	202 (62.2)	326 (63.5)
Utilities	12 (3.7)	28 (5.8)	309 (96.3)	456 (94.2)
Transportation	34 (10.5)	50 (10.3)	290 (89.5)	437 (89.7)
Housing	73 (23.3)	102 (20.8)	240 (76.7)	388 (79.2)
Food	68 (21.6)	99 (19.8)	247 (78.4)	402 (80.2)
Intimate Partner Violence	15 (4.9)	21 (4.4)	289 (95.1)	460 (95.6)
Maternal Depression	20 (27.4)	28 (30.1)	53 (72.6)	65 (69.9)

* NOTE: % is out of total caregiver, or total child.

**Table 3 ijerph-22-01644-t003:** Bi-Variate Analyses Comparing CHILD ACEs Results with Other Psycho-Social Screen Results (N (%) *).

	Positive Child ACEs	Negative Child ACEs
Positive SDOHs (any insecurity)	46 (24.6% had a positive ACE out of those with positive SDOH)	141 (75.4% had a negative ACE out of those with positive SDOH)
Negative SDOHs (any insecurity)	59 (18.1)	267 (81.9)
	*p* = 0.079	
Positive Utilities	7 (25)	21 (75)
Negative Utilities	92 (20.2)	364 (79.8)
	*p* = 0.539	
Positive Transportation	11 (22)	39 (78)
Negative Transportation	89 (20.4)	348 (79.6)
	*p* = 0.786	
Positive Housing	30 (29.4)	72 (70.6)
Negative Housing	71 (18.3)	317 (81.7)
	*p* = 0.014	
Positive Food	20 (20.2)	79 (79.8)
Negative Food	82 (20.4)	320 (79.6)
	*p* = 0.965	
Positive IPV	8 (38.1)	13 (61.9)
Negative IPV	91 (19.8)	369 (80.2)
	*p* = 0.053	
Positive Depression	10 (35.7)	18 (64.3)
Negative Depression	19 (29.2)	46 (70.8)
	*p* = 0.5358	

* NOTE: interpretation of % is written out completely in the first row only.

**Table 4 ijerph-22-01644-t004:** Bi-Variate Analyses Comparing CAREGIVER ACEs Results with Other Psycho-Social Screen Results (N (%)).

	Positive Caregiver ACEs	Negative Caregiver ACEs
Positive SDOHs (any insecurity)	25 (20.3% had a positive ACEs out of those with positive SDOH)	98 (79.7% had a negative ACEs out of those with positive SDOH)
Negative SDOHs (any insecurity)	17 (8.4)	185 (91.6)
	*p* = 0.002	
Positive Utilities	2 (16.7)	10 (83.3)
Negative Utilities	40 (12.9)	269 (87)
	*p* = 0.661	
Positive Transportation	7 (20.6)	27 (79.4)
Negative Transportation	35 (12.1)	255 (87.9)
	*p* = 0.176	
Positive Housing	17 (23.3)	56 (76.7)
Negative Housing	24 (10)	216 (90)
	*p* = 0.003	
Positive Food	12 (17.6)	56 (82.4)
Negative Food	28 (11.3)	219 (88.7)
	*p* = 0.166	
Positive IPV	4 (26.7)	11 (73.3)
Negative IPV	33 (11.4)	256 (88.6)
	*p* = 0.095	
Positive Depression	4 (20)	16 (80)
Negative Depression	10 (18.9)	43 (81.1)
	*p* = 1.0	

NOTE: interpretation of % is written out completely in first row only.

**Table 6 ijerph-22-01644-t006:** Sensitivity Stratified by ACE Exposure Severity.

Number of ACE’s Endorsed	1	2	3	4
Child	44%	50%	73%	67%
Caregiver	40%	44%	53%	60%

## Data Availability

The datasets presented in this article are not readily available because they are derived from information collected as part of routine clinical care. Requests to access the datasets should be directed to Evelyn Berger-Jenkins.

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
