# Peer review of "Adverse Childhood Experiences (ACEs) Screening in Pediatric Primary Care: Is “Social Drivers of Health (SDoH) Screening” Sufficient?"

_ijerph, 2025, doi:10.3390/ijerph22111644_

Round 1
Reviewer 1 Report
Comments and Suggestions for Authors
Overall:
Thank you for the opportunity to review your article. I read the article with great interest, as SDOH screening is an attractive alternative to ACEs screening. The study is simple but effective and adds information to an evolving and important conversation regarding best practices for identifying children and families impacted by ACEs. The study is well-written, the methods are generally appropriate, results are clearly represented, and the discussion is grounded in practical applications for providers and the field.
You do a good job of highlighting the drawbacks and potential harms of ACEs screening in the introduction and teeing up the need for the study. However, I feel like there needs to be a bit more exposition on how the study justified doing ACEs screening in the setting of clear recommendations from AAP/CDC/ACPM, etc. that do not recommend ACEs screening. This might show up in the intro, methods, or discussion, but I just found myself wanting the paper to fill in the dots a bit more. I believe (reading between the lines) that a pilot program had been implemented, and the authors took advantage of the simultaneous administration of the instruments to understand this question… but it felt a bit odd to be reading in the intro about how ACEs screening isn’t recommended and then jump into a paper about how ACEs screening was done.
Maybe a sentence at the end of the first discussion paragraph that says something like, “Researchers have called for additional research on the settings and circumstances in which ACEs screening may be justified, and for alternative screenings or tools that may used in lieu of ACEs screens. In order to understand the comparability of ACEs screening instruments to alternatives, we sought out data from a site that had administered both ACEs and SDOH screening tools.”
Additionally, I think the study could benefit from some sensitivity analyses to understand whether different ACEs cutoffs affect the results. Are SDOH never a good proxy for ACEs? Do results change if you’re looking for high burden (4+) of ACEs among children? Low burden (1+) among adults? I make some specific suggestions below in the methods on the types of sensitivity analyses that would be helpful.
Specific Comments:
Abstract: Appropriate summary of findings. If you have the room, it might be helpful to highlight the NPV in your abstract as well.
Introduction:
- Lines 43-45: the National Survey of Children’s Health is parent-reported, so stating that one third of children aged 0-5 years old endorsed at least one ACE is misleading. Suggest rephrasing to emphasize that the parents are reporting on their children’s experiences (this also clues the reader into the fact that these parent-reported ACEs are likely very underestimated.)
Methods:
- Lines 105-108: please specify the specific tool that was used and consider including the ACEs instrument as a supplemental figure or table
- Did you conduct any sensitivity analyses to examine whether different cutoffs affected your results? While 4+ ACEs is associated with the most outcomes, it is dose-dependent. I think the study could be strengthened by including the results for different ACEs cut points. Are SDOH never a good proxy for ACEs? Do results change if you’re looking for high burden (4+) of ACEs among children? Low burden (1+) among adults? Perhaps Tables 3-5 could have the rows be 4+ | 3+ | 2+ |1+ | 0 ACEs, rather than just ‘positive/negative’. Similarly, Table 6 could show the misclassification rates by level of exposure. Your conclusions will be strengthened if you find that it doesn’t matter what level of ACEs the child/caregiver were exposed to; or, conversely, if there are nuances that need to be understood.
- It’s great that your screening protocol actually led to a more detailed assessment by the trauma treatment program, leading to conversations and referrals that were more therapeutic and helpful than the summary score of ACEs screening; I was concerned that in the quest to answer this question of “can SDOH serve as a proxy for ACEs screening” (which is a necessary question to answer, but difficult since it requires evaluating/comparing to something that is not recommended), the study could potentially cause harm.. but I think the way the ACEs pilot screening handled this was appropriate.
- I think some theoretical explanation for why the combined group was examined is needed – what would be the intervention for a family where there were ACEs among someone, but you don’t know who? It’s the same problem that arises from a score-based screen for very specific ACEs with very specific outcomes and interventions. I think it would be fine to drop the combined group, as your results are likely being driven by caregivers.
- Can you include more information on the pilot program? Were all children 0-5 with a well-visit scheduled automatically included? Were refusals documented? If so, can you include what proportion of well child visits were included in the study?
- How did you define statistical significance? I see that you reference p<0.0533 in the results as statistically significant.
- IRB approval/exemption?
Results:
- Table 1: I think the results are missing for the primary language. It just repeats the names of the languages over and over again.
- Should you suppress cells with less than 10 respondents? Defer to journal policy, but seems potentially identifiable depending on where the study site was.
- The results for IPV and ACEs screening show that you are getting some misclassification on the ACEs screen – all those who answer yes to IPV screen should theoretically be answering yes to at least one of the items on the ACEs screen (I’m assuming you used PEARLS). I would mention this in the discussion.
Discussion:
- Well-written, appropriate scope for results, connects results to real-world implications.
- As I mentioned in the results, the lack of concordance between IPV and ACEs screening shows some misclassification. This should be included in the discussion/limitations.
Author Response
Thank you for your thoughtful review of our manuscript. We very much appreciate all the feedback. Please see the attached document for detailed responses to each comment.

Reviewer 2 Report
Comments and Suggestions for Authors
Abstract
Specify the type of study conducted.
Specify the sample evaluated (mean age and standard deviation) and the instruments used to assess the variables.
Introduction
I find it positive that the authors reviewed and analyzed the difficulties in assessing adverse childhood experiences (ACEs) in primary healthcare settings, given the insufficient evidence and the lack of appropriate protocols to prevent re-victimization.
Another important aspect of this study is that, based on the lack of clarity regarding the assessment of ACEs in primary healthcare, a pilot study of routine ACE screening in pediatric primary care centers was conducted.
Materials and Methods
The methodological aspects of the study are adequately described.
Data Collection and Analysis
The methods were appropriate for addressing the study objective; however, it should be clarified how information was obtained from the instruments used with participants whose first language was Spanish or another language besides English (specify which other languages ​​were spoken by the participants), since this could have affected the reliability of the information and would be a limitation of the study. This data (N%) is not included in Table 1.
It is important to improve the titles of the tables to provide greater clarity regarding the data presented.
Discussion
An interesting contribution of the study is that it found a significant association between positive ACE scores (only in caregivers and in the overall group) and social determinants of health; specifically, housing insecurity. It also found an association between housing insecurity, the presence of ACEs in the child or caregiver, and intimate partner violence. These findings are important for proposing prevention strategies.
Author Response

(The authors gave the same response as above.)
